# Serum Concentrations of Insulin-like Growth Factor-1 (IGF-1), 26S Proteasome (26S PSM), and 3-Methylhistidine (3-MH) in Cats with Hypertrophic Cardiomyopathy

**DOI:** 10.3390/ani15101437

**Published:** 2025-05-15

**Authors:** Stephan Neumann

**Affiliations:** Institute of Veterinary Medicine, Georg-August University of Goettingen, D-37077 Goettingen, Germany; sneuman@gwdg.de

**Keywords:** heart disease, feline, metabolic parameter, IGF-1, 26S PSM, 3-MH

## Abstract

This study investigates the metabolic effects of hypertrophic cardiomyopathy (HCM) in cats (*n* = 27; 13 cats in categories B1 + B2 and 14 cats in categories C + D) by analyzing serum concentrations of insulin-like growth factor-1 (IGF-1), 26S proteasome (26S PSM), and 3-methylhistidine (3-MH). The results demonstrated elevated IGF-1 and 3-MH levels in cats with HCM, with 3-MH showing a tendency to increase in advanced disease stages. These findings indicate a potential association between HCM and altered protein metabolism, underscoring the relevance of metabolic biomarkers in monitoring disease progression.

## 1. Introduction

Cardiac function is a fundamental regulator of various physiological processes, and the systemic metabolic consequences of heart failure, particularly concerning protein metabolism, are well-documented in both human and veterinary medicine [1,2,3,4]. In human patients, impaired cardiac function is associated with disruptions in energy metabolism, which adversely affect disease prognosis [5,6,7]. Clinical manifestations of heart failure commonly include weight loss, muscle wasting, and dysregulation of immune and hormonal functions [5,8]. Similar findings have been reported in veterinary medicine, where cardiac diseases in dogs and cats are linked to alterations in body weight and other clinical abnormalities, primarily driven by metabolic disturbances [2,3,9].

Given the propensity of cardiac disease to induce recognizable metabolic alterations, a detailed examination of these changes is essential. In this study, we analyzed three metabolic biomarkers: IGF-1, 26S PSM, and 3-MH. These biomarkers reflect anabolic and catabolic processes and have been implicated in metabolic dysregulation associated with cardiac dysfunction in humans, dogs, and cats [10,11,12,13,14].

IGF-1 is a polypeptide hormone that plays a central role in the regulation of growth and metabolism, particularly by modulating cell proliferation and differentiation [15]. Serum IGF-1 concentrations have been investigated in various contexts related to protein and carbohydrate metabolism in humans, dogs, and cats [16,17,18]. 26S PSM, a key component of the ubiquitin–proteasome system (UPS), mediates intracellular protein degradation, thereby maintaining protein homeostasis and influencing both anabolic and catabolic pathways [19,20]. Lastly, 3-MH, a post-translationally methylated amino acid derived from myofibrillar proteins, serves as a biomarker of muscle protein degradation. As 3-MH is excreted unchanged, its serum levels provide an index of muscle protein breakdown [21,22,23,24,25].

To elucidate the metabolic effects of cardiac disease and its impact on body condition score (BCS), we investigated cats diagnosed with HCM. We assessed their BCS and measured serum concentrations of IGF-1, 26S PSM, and 3-MH to evaluate potential metabolic alterations.

## 2. Materials and Methods

### 2.1. Study Design

This study included 62 cats in total; 27 belonged to the group diagnosed with HCM and 35 belonged to the healthy control group. The client-owned cats were presented to the Small Animal Clinic of the Institute of Veterinary Medicine, University of Goettingen, Germany between December 2018 and October 2021.

### 2.2. Control Group

The control group comprised cats without any symptoms of respiratory or cardiovascular disease, neither in the case history, nor in physical examination and in a six-month clinical follow up. The cats in this group were examined during routine examinations. A blood analysis including hematology (leucocytes, leucocyte differentiation, packed cell volume, red blood cells, and platelets) and serum chemistry (blood urea nitrogen, creatinine, alanine aminotransferase, alkaline phosphatase, aspartate aminotransferase (AST), total bilirubin, total protein, albumin, globulin, sodium, potassium, calcium, phosphorus, glucose, and cholesterol) was conducted, and only cats without symptoms for disease were included.

### 2.3. HCM-Affected Group (HCM Group)

The cats in the heart disease group were examined either as part of a routine check-up, a follow-up assessment, or due to an acute illness. All cats underwent a comprehensive clinical examination, including blood analysis (hematology and serum chemistry). Cats with comorbidities, e.g., neoplasia, systemic hypertension, or chronic kidney disease, were excluded.

A diagnosis of HCM was established based on echocardiographic findings. This was a diffuse or focal increased left ventricular wall thickness (LVWT; end-diastolic wall thickness ≥ 0.55 cm). To ensure that the increased LVWT was due to HCM rather than other systemic conditions, hyperthyroidism, systemic hypertension, and pseudohypertrophy were systematically ruled out. Furthermore, cats were diagnosed based on the American College of Veterinary Internal Medicine (ACVIM) consensus statement guidelines [26]. The ACVIM stages B1 and B2 were summarized as subclinical stages (B1 + B2), while stages C and D represented stages with previous or acute clinical signs (C + D).

Hyperthyroidism was excluded by measuring thyroxine levels (reference range: 0.9–2.9 µg/dL), with only cats within this range included in the study. Systemic hypertension was ruled out by measuring systemic blood pressure at the time of echocardiography without any anesthesia. Blood pressure was measured using the Doppler oscillometer (VET-HDO, medVet Systems, Im Schloss, Babenhausen, Germany). Based on manufacturer recommendations and the cardiologist’s experience, the threshold value for systemic hypertension was set at >150/95 mmHg. To address potential pseudohypertrophy, the hydration status of each cat was carefully evaluated, with particular attention paid to signs of dehydration.

The body condition score (BCS, scale: 1–9) was evaluated in both groups following the guidelines set by the World Small Animal Veterinary Association (WSAVA) Global Nutrition Committee [27]. All analytical procedures were conducted with prior approval from the regional Consumer Protection and Food Safety authority in Lower Saxony (No 33.9-42502-05-18A361), Germany.

### 2.4. Samples

Blood samples for all subsequent analyses and tests were drawn from the cephalic vein into tubes containing ethylenediaminetetraacetic acid (EDTA) and into serum separator tubes (both supplied by Sarstedt AG & Co. KG, Nümbrecht, Germany). Hematological evaluations were carried out using a ProCyte Dx Analyzer (IDEXX GmbH, Kornwestheim, Germany), while serum biochemical analyses were performed with a Thermo Scientific Konelab 20i Clinical Chemistry Analyzer (Thermo Fisher Scientific Inc., Waltham, MA, USA). For the immunoassays, serum was used (serum tubes; Sarstedt AG & Co. KG, Nümbrecht, Germany). After clotting at room temperature, the serum was centrifuged at 1000× *g* for 20 min (for IGF-1 and 3-MH) or 15 min (for 26S PSM). The serum was then stored in aliquots at –80 °C until analysis.

### 2.5. Echocardiography

Echocardiographic measurements were conducted by a certified veterinary cardiologist. The ultrasonography was performed with the CX50 (Philips, Amsterdam, The Netherlands) using a sector transducer at 4–12 MHz (S12-4, Philips, Amsterdam, The Netherlands). They were part of the usual clinical diagnostic or follow-up examinations. The echocardiographic examination was performed without sedation in lateral recumbency or a standing position. For animals with respiratory distress, the examination was shortened. The linear measurements in both longitudinal and short-axis views encompassed two-dimensional (2D) assessments as well as 2D-guided M-mode imaging. For the purpose of this study, the parameters selected for analysis included left ventricular wall thickness, left atrial size, and the ratio of the left atrium to the aorta. The measurement of left ventricular wall thickness comprised both the left ventricular free wall thickness (LVFWd), assessed in diastole in the right parasternal longitudinal axis, and the interventricular septal thickness (IVSd), likewise measured in diastole in the right parasternal longitudinal axis. In each case, the maximal dimension recorded was included in the analysis. The left atrial size (La) was determined at the end-systole, measured centrally within the left atrium and parallel to the mitral valve plane in the right parasternal longitudinal axis. The ratio of left atrial size (La) to aortic diameter (Ao) was expressed as the dimensionless La/Ao ratio. For this purpose, the left atrium and aorta were visualized in the right parasternal short-axis view at the level of the aortic valve, ensuring the optimal delineation of the left atrium at its maximal expansion.

### 2.6. Assays

Serum concentrations of IGF-1, 26S PSM, and 3-MH were measured using commercially available ELISA kits with different assay procedures. In all three experimental runs, each standard and sample was loaded in duplicate onto the ELISA plate. Serum IGF-1 concentrations (expressed in ng/mL) were determined using a commercially sourced feline sandwich ELISA kit specific for insulin-like growth factor 1 (Cat Insulin-Like Growth Factor 1 ELISA Kit, MBS099428, MyBioSource, Inc., San Diego, CA, USA), comprising 96 wells and offering a quantification range between 6.25 and 200 ng/mL. The manufacturer reported intra-assay and inter-assay coefficients of variation (CV) below 15%. The assay procedure adhered strictly to the manufacturer’s protocol, entailing the sequential addition of standards and serum samples to the antibody-coated microtiter plate, followed by the introduction of an enzyme-conjugated detection reagent. Plates were incubated for 60 min at 37 °C and subsequently subjected to manual washing. A chromogenic substrate mixture, consisting of hydrogen peroxide and tetramethylbenzidine (TMB), was then applied and incubated for approximately 15 min at 37 °C to enable colorimetric signal development. The enzymatic reaction was terminated by the addition of an acidic stop reagent, inducing a color shift from blue to yellow, which facilitated subsequent absorbance measurement.

A commercially available competitive canine 26S PSM ELISA kit (Canine 26S PSM ELISA kit, E08A0669, BlueGene Biotech, Shanghai, China) with 96 wells and a detection range of 0–10 ng/mL was employed to measure serum 26S PSM levels. According to the manufacturer, intra- and inter-assay coefficients of variation (CVs) were reported as <10% and <12%, respectively. The competitive assay was conducted in accordance with the manufacturer’s guidelines, comprising the following procedural steps: Standards and serum samples were dispensed onto a microtiter plate pre-coated with an anti-26S proteasome (PSM) antibody. Subsequently, an enzyme-conjugated detection reagent was added, and the plate was incubated for 60 min at 37 °C, followed by manual washing to remove unbound components. Thereafter, a chromogenic substrate mixture of hydrogen peroxide and tetramethylbenzidine (TMB) was introduced and incubated for approximately 15 min at 37 °C to permit the development of a detectable colorimetric signal. The enzymatic reaction was terminated by the addition of an acidic stop solution, inducing a color transition from blue to yellow, thereby enabling quantitative analysis.

Serum 3-MH concentrations (nmol/mL) were evaluated using a commercially available competitive ELISA kit (3-MH, abx257295, Abbexa Ltd., Cambridge, UK) with general reactivity and 96 wells. The detection range of this kit was indicated to reach from 6.25 to 400 nmol/mL and the intra- and inter-assay CVs were reported to be <8% and <10%, respectively. The competitive ELISA was performed strictly following the manufacturer’s protocol, encompassing the following key steps: The microtiter plate, pre-coated with 3-methylhistidine (3-MH), was first subjected to manual washing. Standards and test samples were then pipetted into the wells, after which a biotin-labeled reagent was added. The plate was incubated at 37 °C for 45 min to allow binding reactions, followed by another manual wash step. Subsequently, an enzyme-conjugated reagent was introduced, and the plate underwent a second incubation phase at 37 °C for 30 min, again followed by washing. The colorimetric reaction was initiated by adding the tetramethylbenzidine (TMB) substrate solution, which was allowed to react for approximately 10 min at 37 °C to develop color. The reaction was halted by introducing an acidic stop solution, triggering a color transition from blue to yellow. Optical density (O.D.) readings were promptly obtained at 450 nm using a TECAN GENios Pro microplate reader (TECAN AUSTRIA GmbH, Grödig, Austria). The quantification of serum concentrations was achieved by constructing a standard curve with CurveExpert Professional 2.6 software (Hyams Development, https://www.curveexpert.net/ (accessed on 1 January 2022)), allowing precise calculation of analyte levels. Importantly, in sandwich-type ELISAs, analyte concentrations are directly proportional to O.D. readings, whereas in competitive assays, an inverse relationship is observed.

### 2.7. Statistical Analysis

Statistical analyses were conducted using GraphPad Prism (version 9.2.0 for Windows, GraphPad Software, San Diego, CA, USA, www.graphpad.com). Descriptive statistics were compiled, and data normality was examined using the D’Agostino–Pearson, Anderson–Darling, and Shapiro–Wilk tests. For group comparisons, non-parametric methods were employed: the Mann–Whitney U test for two-group comparisons, and the Kruskal–Wallis test for comparisons among multiple groups. The following comparisons were made: between the control group and the HCM group, between the control group and various HCM categories, and between the different HCM categories themselves. Boxplots were presented using the Tukey format. Spearman correlation analysis was used to examine the relationships between parameters. Morphological parameters, BCS, and age were treated as continuous variables in the correlation analysis. A *p*-value ≤ 0.05 was considered statistically significant.

## 3. Results

The control group consisted of 35 cats and the HCM group comprised 27 cats. The characteristics of both groups, such as breed, age, sex, neutering status, and BCS, are listed in Table 1. The mean age of the controls was 3.9 years (range 0.5 to 13 years) and the mean age was 8.2 years in the HCM group (range 2 to 14 years). A total of 11 cats in the HCM group were undergoing therapy with the following medications: atenolol (*n* = 6), furosemide (*n* = 5), pimobendan (*n* = 3), imidapril (*n* = 3), clopidogrel (*n* = 2), and acetylsalicylic acid (*n* = 1). All cats were evaluated regarding their hydration status and no signs of dehydration could be found. IGF-1, 26S PSM, and 3-MH were all detectable in serum samples from both healthy cats and casts affected by HCM. The values for IGF-1 ranged from 10.5 to 77.6 ng/mL in the control group and from 27.8 to 59.9 ng/mL in the HCM group; values for 26S PSM ranged from 0.18 ng/mL to 3.58 ng/mL for controls and from 0.20 to 3.49 ng/mL for HCM; and values for 3-MH ranged from 25.1 to 345 nmol/mL for controls and from 112 to 1141 nmol/mL for HCM, respectively. The median values and ranges for both groups are listed in Table 2.

### 3.1. Group Comparisons

All three parameters were compared between the control group and HCM group, with the following results: we found significantly higher serum IGF-1 (*p* < 0.001) and 3-MH concentrations (*p* < 0.001) in the HCM group than in the control group (Figure 1). For 26S PSM, we found no significant difference between the control group and HCM group (*p* = 0.39) (see Figure 1). IGF-1, 26S PSM, and 3-MH were then evaluated separately, regarding several features of cardiac disease (see below). We drew comparisons both between the control group and the disease subgroups, as well as between the disease subgroups themselves. The following parameters were considered: occurrence of increased end-diastolic LVWT (≥0.55 cm), including free wall and septum hypertrophy as well as focal and diffuse hypertrophy; enlargement of La (normal: <1.60 cm, enlarged: ≥1.60 cm); elevated La/Ao (normal: <1.50; elevated: ≥1.50); and the respective disease stage (A–D). Briefly, all 27 cats had a left ventricular wall thickness of > 0.55 cm; 17 cats exhibited an enlarged left atrium; and 17 cats had an increased La/Ao (the La/Ao value was not available for one cat). A total of 13 cats were in category B1 + B2 and 14 cats were in category C + D. The distribution of different morphological parameters in the HCM group, together with the stages according to ACVIM guidelines, are shown in Table 3.

### 3.2. IGF-1: Analysis of Subgroups

For IGF-1, serum concentrations showed a significant difference between the control group and cats in the HCM group, for those without an enlarged atrium (*p* = 0.003) as well as those with an enlarged atrium (*p* < 0.001), with a higher median of IGF-1 serum concentrations in the HCM category in each case (Figure 2, Table 2). No significant difference was found within the HCM group between the categories of cats with and without an enlarged left atrium (*p* = 0.94). The situation was similar for the analyses of La/Ao and staging. The difference in IGF-1 serum concentrations between the control group and cats with HCM without elevated La/Ao (*p* = 0.02) and with elevated La/Ao (*p* < 0.001), respectively, and between the control group and both HCM staging categories, namely B1 + B2 (*p* < 0.001) and C + D (*p* = 0.005), was significant. However, no significant difference was found within the HCM group between elevated and non-elevated La/Ao (*p* = 0.46) or between staging categories B1 + B2 and C + D (*p* = 0.42).

### 3.3. 26S PSM: Analysis of Subgroups

No significant differences were found between the control group and the HCM group nor between HCM categories themselves in the analysis of 26S PSM (Figure 3; for values, see Table 2).

### 3.4. 3-MH: Analysis of Subgroups

For 3-MH, significant differences in serum concentrations were found in the following comparisons: comparison between the control group and cats in the HCM group without an enlarged atrium (*p* < 0.001) as well as those with an enlarged atrium (*p* < 0.001), with a higher median of 3-MH serum concentrations in the HCM group in each case (Figure 4, Table 2); comparisons between the control group and cats in the HCM group without elevated La/Ao (*p* < 0.001) and with elevated La/Ao (*p* < 0.001), respectively; and comparison between the control group and both HCM staging categories, B1 + B2 (*p* < 0.001) and C + D (*p* < 0.001). Any remaining comparisons concerning 3-MH exhibited no significant differences, similarly to the results for IGF-1.

### 3.5. Correlation Analyses

The control group had a mean age of 3.9 years (range: 0.5–13 years), whereas the HCM group had a mean age of 8.2 years (range: 2–14 years). To assess whether IGF-1 levels might be influenced by age-related growth effects, Spearman’s rank correlation was applied to examine the relationship between serum IGF-1 concentrations and age within each group. No significant correlation was observed in either the control group (*p* = 0.97, r = 0.0069) or the HCM group (*p* = 0.78, r = −0.0573), indicating no detectable age effect on IGF-1 levels. Additional correlation analyses using the Spearman method were performed to investigate associations between measured biomarkers and other variables. No significant correlations were identified between 26S proteasome (PSM) or 3-methylhistidine (3-MH) levels and age or body weight. Similarly, no meaningful relationships were found linking serum IGF-1, 26S PSM, or 3-MH concentrations to morphological or clinical parameters, including left ventricular wall thickness (LVWT), left atrial size (La), the La/Ao ratio, ACVIM classification, or body condition score (BCS).

## 4. Discussion

This study investigated serum concentrations of three metabolites in cats affected by HCM and in healthy controls, correlating these measurements with morphological echocardiographic parameters and body condition scores (BCS). HCM is a cardiomyopathy characterized by structural myocardial impairment, which, beyond affecting the cardiovascular system, has systemic implications, particularly in the context of chronic heart failure (CHF). The metabolic impact of HCM can be assessed using physical condition parameters such as weight, BCS, or muscle condition score (MCS) [2,5,28]. In the present study, BCS distribution was comparable between the study groups (the median BCS was five in both groups; see Table 1), with sporadic higher values in the HCM group. Further insights can be gained at the molecular level through metabolite quantification [1,6]. Given the promising outcomes of prior studies utilizing multiple metabolic parameters in cardiovascular research [29], we sought to achieve a detailed understanding by analyzing IGF-1, 26S PSM, and 3-MH.

### 4.1. IGF-1

Several previous studies have examined IGF-1 in cats [11,18,30]. Our findings regarding IGF-1 concentrations align with prior research indicating higher median IGF-1 levels in cats with HCM compared to healthy controls [11,30], although one study reported no significant difference between these groups [31]. A study in human patients with HCM found significantly elevated IGF-1 concentrations in individuals with hypertrophic non-obstructive cardiomyopathy (HNCM) and hypertrophic obstructive cardiomyopathy (HOCM), whereas IGF-1 levels were lower in patients with HCM complicated by CHF compared to healthy controls [32]. Consistently, our study revealed that cats classified in stages C + D had a tendency to lower median IGF-1 concentrations than those in stages B1 + B2 (Table 2). These findings are consistent with those of Saeki et al. [32], who reported that IGF-1 levels in patients with CHF were lower than those in healthy individuals, as well as those in other HCM subgroups. Elevated IGF-1 concentrations in HCM are considered indicative of cell maturation and myocardial growth responses [11], playing a role in disease progression. Conversely, decreased IGF-1 levels in advanced-stage HCM may be associated with progressive myocardial damage [32]. However, the causal relationship between IGF-1 concentrations, myocardial damage, and cardiac function remains undetermined.

### 4.2. 26S-PSM

The ubiquitin–proteasome system (UPS) is implicated in chronic diseases, yet findings regarding its role in cardiomyopathies remain inconsistent. In our study, 26S PSM serum concentrations did not differ significantly between control and HCM-affected cats. As a key component of the UPS, 26S PSM is involved in intracellular protein degradation and has been associated with proteolysis in human cardiomyopathies. Vosberg [33] highlighted its role in protein turnover and suggested its potential contribution to HCM pathogenesis via disease-related disruptions in protein degradation. Several studies have demonstrated links between the UPS and human cardiomyopathies [12,34,35]. Despite left ventricular hypertrophy being a hallmark of HCM, we found no significant differences in serum 26S PSM concentrations between affected and control cats. Methodological considerations may account for this, as different measurement approaches—ranging from proteasome content and activity analyses to investigations of specific subunits—have yielded varying results [36,37,38]. Species differences were ruled out as a confounding factor, given the high sequence homology (95–100%) of 26S PSM subunits between *Canis lupus familiaris* and *Felis catus*, as confirmed by UniProt sequence alignments. Another possible explanation for the lack of significant findings is the comparable BCS between study groups, which contrasts with previous findings showing elevated 26S PSM concentrations in dogs with chronic diseases and reduced BCS [36].

### 4.3. 3-MH

Elevated serum 3-MH concentrations in HCM-affected cats in this study are consistent with prior veterinary research, which reported significantly higher 3-MH levels in dogs with chronic mitral valvular disease and heart failure compared to healthy controls [14,29]. These studies, however, employed alternative quantification methods rather than ELISA assays. Increased 3-MH concentrations have also been observed in other veterinary contexts, including liver and neoplastic diseases in feline and canine populations [13,39].

Although no significant differences in 3-MH levels were detected across HCM severity categories—whether based on left atrial size, La/Ao ratio, or clinical staging (B1 + B2 vs. C + D)—a trend was observed: unlike IGF-1, median 3-MH concentrations were higher in advanced disease stages (C + D) than in earlier stages (B1 + B2). This trend is consistent with findings in canine mitral valve disease, where elevated 3-MH levels were noted in the most severe heart failure stage (International Small Animal Cardiac Health Council class III) but not in intermediate stages [14]. Although our study did not establish correlations between 3-MH and echocardiographic parameters, Li et al. [29] identified a significant positive correlation between left atrial diameter and 3-MH in dogs with myxomatous mitral valve disease. Given its established association with increased myofibrillar turnover and degradation [13,14], 3-MH emerges as a promising biomarker for myocardial stress and remodeling in HCM. These findings underscore the need for further investigation into its role in the pathophysiology of cardiac disease, particularly regarding its utility for early detection and disease monitoring.

### 4.4. Limitations

This study provides valuable insights despite certain limitations. As a clinical study involving outpatient cats, we could not standardize dietary intake or measure energy consumption. Although age-related influences on biomarker measurements were considered, echocardiography and the measurement of blood pressure were not performed on control cats; however, the absence of heart murmurs and gallop rhythm or clinical symptoms minimizes potential misclassification. In a recently published study, clinical parameters from 354 cats were investigated to predict HCM. Missing heart murmur and gallop rhythms were used to correctly predict 90% of healthy cats [40].

Biomarker stability was maintained despite a storage period of approximately three months. As feline-specific ELISA kits for 26S PSM and 3-MH were unavailable, we employed validated mammalian assays based on homology research, which yielded comparable results. Notably, 26S PSM concentrations in our control group corresponded to findings from a canine study using the same assay [36]. Similarly, 3-MH concentrations in our study mirrored those of [14], demonstrating a comparable threefold increase in cardiac disease cases and thereby reinforcing data reliability.

While peripheral metabolite measurements do not directly assess cardiac tissue dynamics, prior research [37] suggests a strong link between systemic proteasomal activity and cardiac conditions. Although proteasomal expression and activity may differ [10], serum 26S PSM quantification provides meaningful insights into proteasomal function in feline cardiac disease.

Despite methodological constraints, this study advances our understanding of cardiac biomarkers in cats and highlights their potential diagnostic relevance. Of the 27 cats in the HCM group, 11 were undergoing therapy with the following medications: atenolol (*n* = 6), furosemide (*n* = 5), pimobendan (*n* = 3), imidapril (*n* = 3), clopidogrel (*n* = 2), and acetylsalicylic acid (*n* = 1). No direct influences of these drugs on the measured biomarkers could be found in the literature. An indirect influence of furosemide is possible, because it affects the hydration status. We tried to exclude this influence by carefully observing the cats for signs of dehydration.

## 5. Conclusions

The results for the serum parameters investigated here imply a possible interaction between HCM and protein turnover in cats. According to our results, it is not possible to use these three parameters to distinguish between different degrees of HCM. While our data on serum IGF-1 and 3-MH levels in HCM-affected cats are largely similar to what has been previously found in studies in human and veterinary medicine on these markers in the context of heart disease, the role of 26S PSM could not be assessed conclusively in this context. Examining several parameters or even larger profiles of metabolites as an approach to assessing their interplay, as well as the effects of diseases on the whole body, seems promising to us for further analyses.

## Figures and Tables

**Figure 1 animals-15-01437-f001:**
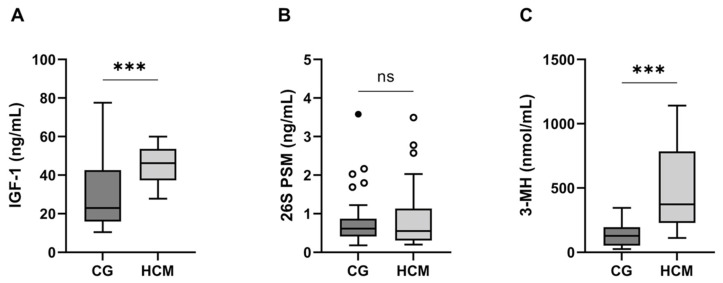
Comparison of IGF-1 (**A**), 26S PSM (**B**), and 3-MH (**C**) between the HCM group and the control group. CG: control group; HCM: HCM group; ○: outlier; •: extreme value. *** *p* < 0.0001; ns: not significant.

**Figure 2 animals-15-01437-f002:**
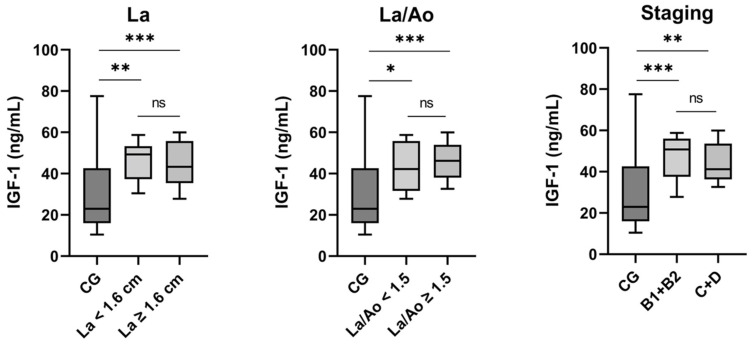
Comparisons of IGF-1 concentrations for control group and HCM subgroups (left atrium (La), left atrium/aorta (La/Ao), Staging). CG: control group. * *p* < 0.05, ** *p* < 0.001, *** *p* < 0.0001, ns: not significant.

**Figure 3 animals-15-01437-f003:**
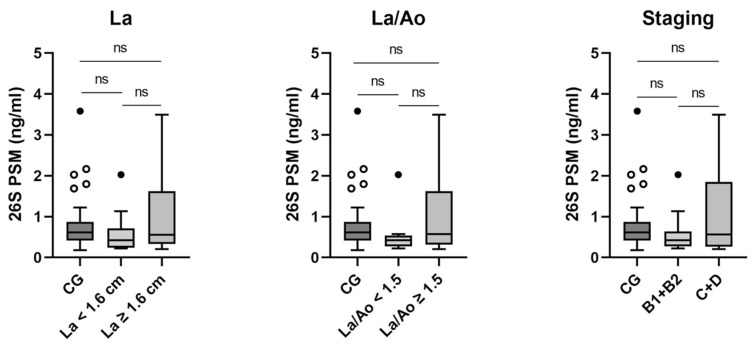
26S PSM in the control group and HCM subgroups (left atrium (La), left atrium/aorta (La/Ao), Staging). CG: control group; ○: outlier; •: extreme value. ns: not significant.

**Figure 4 animals-15-01437-f004:**
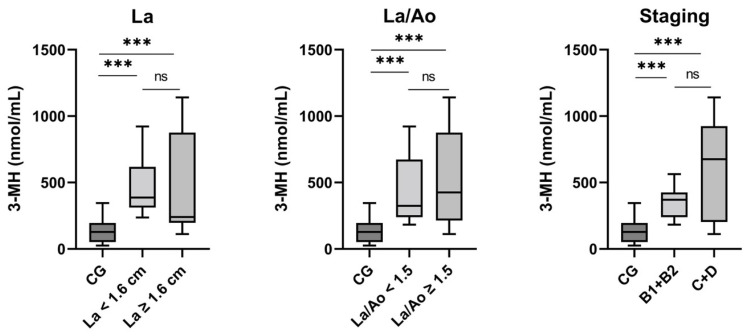
3-MH of the controls and HCM subgroups (left atrium (La), left atrium/aorta (La/Ao), Staging). CG: control group. *** *p* < 0.0001, ns: not significant.

**Table 1 animals-15-01437-t001:** Number, breed, age, sex, neutering status and BCS in control group and HCM group.

	Control Group	HCM Group
n total	35	27
Breed	n	n
Bengal	-	2
Birman	1	-
British Shorthair (and crossbreeds)	1	4
European Shorthair	28	15
Maine Coon	1	2
Norwegian Forest Cat	1	-
Persian (and crossbreeds)	1	2
Scottish Fold	-	1
Siamese (and crossbreeds)	2	-
Sphynx	-	1
Age (years)	n	n
0–5	25	9
6–10	7	12
11–15	3	6
Sex and neutering status	n	n
female (intact/neutered)	14 (9/5)	7 (1/6)
male (intact/neutered)	21 (8/13)	20 (4/16)
BCS (score 1–9)	n	n
3	-	1
4	11	5
5	18	15
6	6	3
7	-	3
Body weight (kg)	n	n
3–4	7	7
4–5	23	17
5–6	4	3
>6	1	0

BCS: body condition score.

**Table 2 animals-15-01437-t002:** Median and range for IGF-1, 26S PSM and 3-MH concentrations in control group and HCM group and in HCM subgroups (med [min; max]).

	IGF-1 (ng/mL)
Control group	23.0 [10.5; 77.6]
HCM group	46.2 [27.8; 59.9]
La < 1.60 cm	49.3 [30.4; 58.7]
La ≥ 1.60 cm	43.3 [27.8; 59.9]
La/Ao < 1.50	42.2 [27.8; 58.7]
La/Ao ≥ 1.50	46.2 [32.6; 59.9]
Staging B1 + B2	50.8 [27.8; 58.8]
Staging C + D	41.2 [32.6; 59.9]
	26S PSM (ng/mL)
Control group	0.61 [0.18; 3.58]
HCM group	0.55 [0.20; 3.49]
La < 1.60 cm	0.42 [0.22; 2.03]
La ≥ 1.60 cm	0.55 [0.20; 3.49]
La/Ao < 1.50	0.42 [0.22; 2.03]
La/Ao ≥ 1.50	0.57 [0.20; 3.49]
Staging B1 + B2	0.42 [0.22; 2.03]
Staging C + D	0.56 [0.20; 3.49]
	3-MH (nmol/mL)
Control group	128 [25.1; 345]
HCM group	373 [229; 1141]
La < 1.60 cm	387 [237; 922]
La ≥ 1.60 cm	240 [112; 1141]
La/Ao < 1.50	324 [183; 922]
La/Ao ≥ 1.50	426 [112; 1141]
Staging B1 + B2	370 [183; 563]
Staging C + D	675 [112; 1141]

La left atrium, La/Ao left atrium/aorta, med median, min minimum, max maximum.

**Table 3 animals-15-01437-t003:** Distribution of morphological parameters in the HCM group.

Left ventricular wall (cm)	n
0.55–0.64	4
0.65–0.70	3
>0.70	20
Left atrium (cm)	n
<1.60	10
≥1.60–1.90	7
>1.90	10
La/Ao	n
<1.50	9
≥1.50–1.80	5
>1.80	12
Not applicable	1
ACVIM stage	n
A	-
B1	8
B2	5
C	13
D	1

La/Ao left atrium/aorta.

## Data Availability

No new data were created or analyzed in this study.

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
