# Peer review of "Serum Concentrations of Insulin-like Growth Factor-1 (IGF-1), 26S Proteasome (26S PSM), and 3-Methylhistidine (3-MH) in Cats with Hypertrophic Cardiomyopathy"

_animals, 2025, doi:10.3390/ani15101437_

Round 1

Reviewer 1 Report

Comments and Suggestions for Authors

The author evaluated serum levels of IGF-1, 26S PSM, and 3-MH in healthy control cats (n = 35) and cats diagnosed with hypertrophic cardiomyopathy (HCM) (n = 27). While the study presents a valuable contribution to the field, several points require clarification.

  • Abbreviations: From the moment an abbreviation is introduced, it must be used in the text (e.g. hypertrophic cardiomyopathy). Also, it should be avoided to start sentences with abbreviation.
  • Title: The title should be revised to make the message more concise and focused.
  • In both the summary and the abstract, the staging of cats with HCM should be included.
  • Abstract_Lines 23-24: The statistical analysis did not demonstrate a significant difference in 3-MH levels between cats in stages C/D and those in stages B1/B2. This sentence should be rephrased.
  • In the Introduction and Discussion, when the author refers to previous studies with these biomarkers in the context of cardiovascular diseases, it should always be specified whether these studies were carried out in humans or in animals (experimental or clinical studies), and in which species (e.g. lines 282-283: there are more examples throughout the text). In the case of IGF-1, there are several previous studies in which this biomarker has been evaluated in cats with HCM, so the author should cite all these articles and mention how the present study complements the information already available in the literature.
  • Line 72: What serum biochemistry analyses were carried out on the control cats?
  • Did cats with HCM have other comorbidities? If yes, they should be mentioned.
  • Lines 86-87: The technique of measuring arterial pressures in cats with HCM should be included.
  • Lines 123-127: This information should be included in the section describing the group of cats with HCM, as these guidelines consider clinical parameters in addition to echocardiographic ones.
  • Regarding the echocardiographic diagnosis of HCM, were there any cases with left ventricular outflow tract obstruction?
  • Groups were not age-matched (healthy: 3.9 years; HMC: 8.2 years), which could bias the obtained results – this should be discussed. Another aspect relates to body weight, which should also be included in the results, in addition to body condition score.
  • Table 2: Consider presenting the table with a horizontal organization.
  • Lines 269-270: Regarding the correlation analyses, how did 26S PSM and 3-MH correlate with age and how did all the biomarkers correlate with body weight?
  • Lines 294-296: These results were not statistically significant. This information should be considered in the discussion.
  • Lines 302-304: explain why the correlaion between IGF-1 levels and the age of cats was carried out, based on existing literature.
  • Lines 325-327: In cats with HCM, was echocardiographic assessment of systolic function carried out? In this passage of the text, it would be interesting to include the human medicine article (“Targeted Quantitative Plasma Metabolomics Identifies Metabolite Signatures that Distinguish Heart Failure with Reduced and Preserved Ejection Fraction”), in which no differences in serum 3-MH levels were observed between heart failure patients with preserved ejection fraction and patients with reduced ejection fraction, and to draw possible parallels with diseases in the field of veterinary medicine.
  • Lines 331-333: This information repeats the message on lines 325-328.
  • Line 355: In the discussion, one of the limitations mentioned is the fact that the control cats were not subjected to echocardiographic assessment. Despite this, the author states that as the cats had no murmurs or clinical signs, it supports the idea that they were healthy cats. This statement should be supported by literature on the signs of cardiomyopathy in asymptomatic cats.

Author Response

Dear reviewer,

thanks for your support and the chance to improve my manuscript. I have attached your comments with my answered.

Best regards

Stephan Neumann

Reviewer 2 Report

Comments and Suggestions for Authors

I thank the authors for conducting this important study on various biomarkers in control cats and those with HCM-phenotype cardiomyopathy. The investigation into the correlation and differences in biomarker levels between groups provides further insight into the pathophysiology of feline HCM.

However, a major limitation of the current study is the lack of echocardiographic screening in the control group. Without echocardiography, it is not possible to definitively rule out HCM-phenotype cardiomyopathy in these cats, potentially confounding group comparisons and weakening conclusions.

Below are some concerns and suggestions that I believe, if addressed, could enhance the manuscript:

Line 68

Regarding the control group: was echocardiography performed as part of screening to rule out subclinical cardiomyopathy or other cardiac conditions that may affect the study design?

Additionally, were systemic hypertension and hyperthyroidism excluded in these cats? As both conditions can be subclinical and may influence cardiac structure and function—as well as the measured biomarkers—clarification on how these were assessed would be valuable.

Line 87

Please elaborate on the cut-off used to define systemic hypertension. Were cats conscious, sedated, or under tranquilization during blood pressure measurement? What method was used (e.g., Doppler, oscillometric)? Was the measurement taken before or after echocardiography, and was the timing standardized?

Line 117

Please specify whether the analysis used the highest value from LVFW or IVS, and whether measurements were obtained via M-mode or 2D imaging. A clearer methodological description would benefit the readers.

Line 119

A more detailed explanation—or even an illustration—of LA measurement methods and LA/Ao ratio assessment would enrich the manuscript. How was LA diameter measured in the right parasternal long-axis view (e.g., perpendicular to the mitral annulus)? At which phase of the cardiac cycle was LA/Ao measured, and how were Ao and LA defined in the short-axis view?

Line 192

Table 1 suggests a notable age difference between control and HCM cats. Was this statistically significant? Since younger age is common in control groups, clarification would help readers interpret group differences.

Also, considering comparisons are made between HCM stages B1/B2 and C/D, is there a significant age difference between these subgroups? Including this in Table 1 would be informative.

Line 216

Were there cats with borderline LA enlargement (e.g., increased LA in long-axis but normal LA/Ao, or vice versa)? If so, how was LA enlargement defined? Among the 17 cats with increased LA/Ao, how many also had LA enlargement in the long-axis view?

Line 218

How was Stage D HCM defined in the single affected cat—based on diuretic dosage or other criteria? What parameters were used to differentiate Stage B1 from B2?

Line 269

Was correlation analysis performed between IGF-1, 26S PSM, and 3-MH? Also, were these results derived only from the HCM group? If control cats did not undergo echocardiography, this should be clarified in the text to contextualize the analysis.

Line 293

Although median IGF-1 values appear lower in CHF cats, the range and boxplot suggest overlap. Rather than stating a decrease, it may be more appropriate to say levels were similar. Given only one cat was in Stage D, it remains unclear whether a larger number of end-stage cases would yield more distinct results.

Discussion

Several cats received treatment for HCM, including furosemide for CHF. While cats with CKD were excluded, the potential effects of treatment on biomarker levels should be discussed, as treatment could influence study outcomes.

Line 354

I appreciate the rationale for omitting echocardiography in the control group. However, it should be acknowledged that misclassification is possible, as asymptomatic cats can have undiagnosed HCM. Additionally, auscultation is not a reliable method to rule out cardiomyopathy.

Was systemic hypertension also evaluated in the control group? If not, some cats with hypertensive cardiac target organ damage may have been misclassified, potentially affecting biomarker interpretation.

Author Response

(The authors gave the same response as above.)

Round 2

Reviewer 1 Report

Comments and Suggestions for Authors

Dear author,

Avoid using abbreviations at the beginning of sentences. Do not use the abbreviation HCM in the title. Lines 94-96: insert bibliographic reference supporting this information

Author Response

Dear reviewer,

I have changed the text based on your remarks.

Best regards

Stephan Neumann

Reviewer 2 Report

Comments and Suggestions for Authors

I have no further question

Author Response

Thank you for your review.

Best regards

Stephan Neumann